# Risk Factors and Long-Term Outcomes of Acute Kidney Disease in Hematopoietic Stem Cell Transplant—Cohort Study

**DOI:** 10.3390/cancers17030538

**Published:** 2025-02-05

**Authors:** Natacha Rodrigues, Carolina Branco, Gonçalo Sousa, Manuel Silva, Cláudia Costa, Filipe Marques, Pedro Vasconcelos, Carlos Martins, José António Lopes

**Affiliations:** 1Division of Nephrology and Renal Transplantation, Unidade Local de Saúde Santa Maria, 1649-035 Lisbon, Portugal; carolina.branco@chln.min-saude.pt (C.B.); claudia.r.s.costa@chln.min-saude.pt (C.C.); filipedcmarques@campus.ul.pt (F.M.); jose.lopes@chln.min-saude.pt (J.A.L.); 2Faculdade de Medicina da Universidade de Lisboa, 1649-035 Lisbon, Portugal; goncalo-sousa@edu.ulisboa.pt; 3Division of Haematology, Unidade Local de Saúde Santa Maria, 1649-035 Lisbon, Portugal; pedro.de.vasconcelos.monteiro@gmail.com (P.V.); carlos.martins@chln.min-saude.pt (C.M.)

**Keywords:** acute kidney disease, acute kidney injury, hematopoietic stem cell transplant, outcomes

## Abstract

The increasing demand for hematopoietic stem cell transplants underscores the need for a deeper understanding of associated complications, particularly renal complications which have been consistently linked to this procedure. While acute kidney injury has been extensively studied in the context of this procedure, data regarding acute kidney disease and its long-term implications remain scarce. We present a clinical study exploring the prevalence and implications of acute kidney disease in patients undergoing this procedure. Our study enrolled 422 patients, and we present our findings on cumulative incidence, risk factors, and long-term outcomes, which address a significant gap in the current literature.

## 1. Introduction

The role of hematopoietic stem cell transplant (HSCT) has gained increasing importance in clinical practice as hematologic malignancies attain higher diagnostic rates [1]. The prognostic impact of this procedure is nowadays unquestionable as it represents a potentially curative treatment for virtually all hematologic cancers. Acute leukemias are the most common indication for allogeneic HSCTs, and multiple myeloma is the most common adult indication for autologous HSCTs followed by Relapsed/refractory Hodgkin (HL) and non-Hodgkin lymphomas (NHL) [2].

Activity reports from the Center for International Blood and Marrow Transplant Research (CIBMTR) demonstrate increasing numbers of total HSCTs performed each year with improving survival over time [3]. The expanded use of both autologous and allogeneic HSCT for patients of older ages is thought to be related to improved supportive care and the effective use of reduced-intensity conditioning regimens for patients ineligible for myeloablative therapy [4]. With the rising demand for HSCT, the global number of patients undergoing this procedure has increased by approximately 7% annually over the past five years, a trend that is projected to persist in the coming years [5]. This scenario highlights the pressing need to deepen our understanding of HSCT-related complications, particularly the kidney dysfunctions that frequently accompany this procedure.

Acute kidney injury (AKI) is a well-characterized entity that has been vastly studied in HSCT recipients in the last decade, complicating 45 to 65% of procedures and being associated with lower overall survival in higher stages of AKI severity [6,7,8,9].

By definition, AKI lasts for ≤7 days [10]. There is a considerable number of patients who develop persistent kidney damage, with delayed recovery, during the course of the disease. Acute kidney disease (AKD) has been proposed to define the course of disease after AKI among patients in whom the renal pathophysiologic processes are ongoing and last for ≤3 months [11]. A comprehensive systematic review has shown that 26.11% of patients developed AKD following an episode of AKI [12]. Mima et al. 2019 published a study involving a cohort of 108 HSCT patients, showing an AKD incidence of 15.7% and considering only serum creatinine criteria [13]. Research considering different clinical settings consistently showed that AKD is linked to a poorer prognosis (including higher risks of chronic kidney disease (CKD) and mortality). These include hospitalized individuals [14], surgical patients [15], and those with acute decompensated heart failure [16]. To our knowledge, there is no epidemiologic information on AKD incidence considering both creatinine and urinary output criteria, nor data on the long-term outcomes of AKD in the HSCT population.

An awareness of the risk factors for AKD allows for the identification of high-risk patients, enabling the timely implementation of preventive measures to alleviate the progression and impact of the disease.

Our study aims to determine the cumulative incidence of AKD in the first 100 days after HSCT; to identify independent risk factors for AKD in HSCT; and to determine the impact of AKD in 3-year overall survival and relapse-free survival in HSCT.

## 2. Materials and Methods

### 2.1. Study Design, Population, and Data Collection

We conducted a single-centre retrospective cohort study. We included patients with hematological malignancies—leukemia, lymphoma, multiple myeloma—admitted for HSCT at Unidade Local de Saúde Santa Maria (ULS-SM) between January 2005 and December 2015. We included patients undergoing auto-HSCT and allo-HSCT with myeloablative and reduced-intensity conditioning (RIC) treatments.

We did not include patients under the age of 18 years; patients with a history of autologous or allogeneic HSCT; patients with chronic kidney disease already receiving renal replacement therapy, and patients who had undergone renal replacement therapy within a week prior to transplantation.

The conditioning regimens used were defined according to institutional guidelines and the patient’s clinical profile as well as the specific hematologic diagnosis. Total body irradiation is not only unavailable at ULS-S but also not present in any institutional protocol, at ULS-SM. Prophylactic antimicrobial therapy consisted of co-trimoxazole, ciprofloxacin, acyclovir and fluconazole.

We collected data from records of HSCT eligibility assessments, daily medical notes, six-hour-period nurse records, and diagnostic exams conducted throughout the hospital stay for HSCT, in all appointments and hospital admissions in the first 100 days after HSCT; medical records and laboratory analysis were carried out in the first three years after HSCT.

We collected data on various patient variables such as demographic characteristics and medical history. Additionally, we recorded information on the hematologic diagnosis and HSCT-related factors. Moreover, data on disease relapse timing and overall survival duration were also collected.

Patients had follow-up until death or until they reached the censoring point, which was set at 36 months (3 years) post-HSCT. This timeframe was chosen since, beyond this period, patients are typically referred to other hospitals closer to their residences.

### 2.2. Definitions

The baseline serum creatinine level was defined as the value recorded upon hospital admission before the conditioning regimen. The baseline glomerular filtration rate was calculated using the CKD-EPI equation [17], based on the previously defined baseline serum creatinine level.

Chronic kidney disease (CKD) was classified following KDIGO guidelines [18], i.e., as a sustained reduction in estimated glomerular filtration rate to below 60 mL/min/1.73 m^2^ for a period exceeding three months.

AKI was defined using the KDIGO clinical practice criteria, which include any of the following: a rise in serum creatinine of at least 0.3 mg/dL (26.5 μmol/L) within 48 h; an increase in serum creatinine to at least 1.5 times the baseline level, occurring or suspected to have occurred within the past seven days; or a urine output of less than 0.5 mL/kg/h for six hours. These criteria were assessed using daily serum creatinine values and six-hour urinary output measurements until hospital discharge, as well as during any subsequent hospital admissions or weekly outpatient clinic visits for the first 100 days following HSCT.

AKD was established when AKI was present, and the patient continued in KDIGO stage 1 or higher after 7 days [19].

The hematopoietic cell transplantation-specific comorbidity index (HCT-CI) was calculated using a 2005 validated version [20].

When referring to nephrotoxic drugs, we considered the administration of amikacin, vancomycin, gentamicin, foscarnet, and amphotericin B.

Relapse-free survival was calculated in days from HSCT until disease relapse.

Overall survival was calculated in days from HSCT until any cause of death.

### 2.3. Statistical Methods

Categorical variables were presented as frequencies (percentages), while quantitative data were described as median (P25 = 25th percentile; P75 = 75th percentile). The primary outcomes were the cumulative incidence of AKD, disease-free survival, and overall survival. For the first two outcomes, statistical methods recommended by the European Group for Blood and Marrow Transplantation [21] were applied, specifically survival analysis techniques that account for competing events using the Fine and Gray method [22]. In this framework, death was treated as a competing event in both univariable and multivariable analyses to identify factors contributing to AKD risk and evaluate its impact on disease-free survival. Additive Cox proportional hazards regression models were used to analyze the time to death from all causes.

To develop the final multivariable model, stepwise selection regression techniques were employed. The assumption of Cox proportional hazards was verified using both formal statistical tests and graphical assessments based on scaled Schoenfeld residuals. Hazard ratios, both crude and adjusted, were calculated along with their 95% confidence intervals (CIs). A significance level of α = 0.05 was applied. Data analysis was conducted using STATA for Windows (StataCorp, 2019. Stata Statistical Software: Release 16, College Station, TX, USA: StataCorp LLC.) and R software version 2017 (R Core Team (2017)). R: A language and environment for statistical computing, Foundation for Statistical Computing, Vienna, Austria. URL https://www.R-project.org/ accessed on 9 September 2024).

## 3. Results

Between January 2005 and December 2015, five hundred and thirty-four patients underwent HSCT in our center. Among these patients, one hundred and twelve had at least one exclusion criteria and 422 patients were eligible for the study. Demographic and clinical patients’ characteristics are shown in Table 1.

### 3.1. Cumulative Incidence of AKD

In the first 100 days after HSCT, the cumulative incidence of AKI was 59.1% and the cumulative incidence of AKD was 22.9% (95% CI: 19.2–27.4%)—Figure 1.

### 3.2. Risk Factors Associated with Acute Kidney Disease

The univariable analysis for AKD considering death as a competing risk is presented in Table 2. In this analysis variables associated with AKD were BMI (HR: 1.04, 95% CI 1.00–1.08; *p* = 0.036) HCT-CI score ≥ 2 (HR: 1.65, 95% CI 1.02–2.67; *p* = 0.040), hypertension (HR: 1.70, 95% CI 1.10–2.63; *p* = 0.016), chronic kidney disease (HR: 2.14, 95% CI1.08–4.23; *p* = 0.029), allogeneic transplant (HR: 1.79, 95% CI 1.20–2.70; *p* = 0.004), shock (HR: 2.14, 95% CI 1.31–3.50; *p* = 0.002), nephrotoxic drugs (HR: 8.33, 95% CI 1.11–62.2; *p* = 0.039), sepsis (HR: 1.53, 95% CI 1.02–2.30; *p* = 0.042) and Reactive C Protein (HR: 1.01, 95% CI 1.00–1.01; *p* < 0.001).

Variables independently associated with a higher incidence of AKD are shown in Table 3 and include the following: BMI (HR: 1.05, 95% CI 1.01–1.10; *p* = 0.034), HCT-CI score ≥ 2 (HR: 1.83, 95% CI 1.11–3.13; *p* = 0.027) allogeneic transplant (HR: 2.03, 95% CI 1.26–3.33; *p* = 0.004), C—Reactive Protein (HR: 1.01, 95% CI 1.01–1.02; *p* < 0.001). Nephrotoxic drugs (HR: 4.81, 95% CI 1.54–4.95; *p* = 0.038).

### 3.3. Prognostic Impact of AKD in Overall Survival

By the end of the 3-year follow-up period, 181 (43.3%) patients had died—64.9% of patients with AKD versus 35.4% of patients without AKD. Overall survival in patients who had an AKD episode in the first 100 days after HSCT was lower compared to those who did not (Log-rank test for equality of survivor functions *p* < 0.001)—Figure 2.

In univariable analysis, variables with an impact on lower overall survival were BMI (HR: 0.94; 95% CI 0.91–0.97; *p*  = 0.001) hypertension (HR: 1.56; 95% CI 1.05–2.32; *p* = 0.029) type of HSCT (reference category allogeneic HR: 3.17; 95% CI 2.36–4.27; *p* < 0.001), platelets at hospital admission (HR: 0.99; 95% CI 0.98 –0.99; *p* = 0.01), AKI (HR: 1.68; 95% CI 1.23–2.30; *p* = 0.001), AKD (HR: 2.27; 95% CI 1.66–3.09; *p* < 0.001), sepsis (HR: 1.63; 95% CI 1.21–2.20; *p* = 0.001), and relapse (HR: 2.21; 95% CI 1.65–2.96; *p* < 0.001)—Table 4.

In multivariable analysis, the variables with independent association with lower overall survival were AKD (HR: 1.75; 95% CI 1.27–2.39; *p* = 0.001), sepsis (HR: 1.49; 95% CI 1.10–2.02; *p* = 0.011), allogenic transplantation (HR: 3.33; 95% CI 2.43 –4.54; *p* < 0.001), relapse (HR: 2.61; 95% CI 1.93–3.54; *p* < 0.001) serum lactate dehydrogenase (HR for each 100 units/L increment: 1.10, 95% CI: 1.03–1.80; *p* = 0.001)—Table 5.

### 3.4. Prognostic Impact of AKD in Disease-Free Overall Survival

Cumulative incidence of relapse was 45.2% at three years after HSCT—39.9% of patients with AKD versus 53.1% of patients with no AKD, with no statistically significant difference amongst groups (Log-rank test for equality of survivor functions *p* = 0.062).

No statistically significant association was found between AKD with lower disease-free overall survival.

### 3.5. Progression from AKD to CKD

Considering the 94 patients that developed AKD within the first 100 days post-transplant, 26 died in the first 3 months after the AKD diagnosis. Among the 68 patients that survived more than 3 months after the AKD diagnosis, 19.1% (13 patients) progressed to CKD.

## 4. Discussion

The concept of AKD was first introduced in 2012 by the KDIGO working group [10] to describe impaired kidney function, identified through either serum creatinine levels or estimated glomerular filtration rate (eGFR), lasting for less than three months. In 2017, the Acute Dialysis Quality Initiative 16 report proposed an alternative definition and classification system for AKD [19], incorporating a framework that categorized AKD based on the severity of preceding AKI. In 2020, KDIGO held a consensus conference on AKD, highlighting its association with increased risks of mortality and the development of chronic kidney disease (CKD) [11]. AKD can be considered a transitional phase between AKI and CKD, representing a crucial period during which medical interventions may influence the progression of kidney disease.

In this study, we investigated the occurrence of AKD (presenting at least KDIGO Stage 1 criteria for >7 days after an AKI-initiating event) in a cohort of 422 patients undergoing HSCT. The incidence of AKD was 22.9%. AKD was independently associated with a lower overall survival conferring a 1.75-fold higher risk of death than patients with no AKD.

In 2019, Mima et al. published a cohort of 108 HSCT patients showing an AKD incidence of 15.7%, considering only serum creatinine criteria and not considering death as a competing event in their analysis [13]. We believe that the inclusion of urinary output criteria in our study allowed us to diagnose more patients with this condition. Also, statistically approaching AKD using survival analysis considering death as a competing event is important in this population given the considerable deaths occurring short term after HSCT. Our results are more consistent with the systematic review and meta-analysis published by Su et al. 2022 showing an incidence rate of hospital-acquired AKD following an AKI episode of 26.11% [12].

Based on our results, the most crucial factors associated with AKD are a higher BMI, an HCT-CI score ≥ 2, an allogeneic HSCT, a higher C-reactive protein at admission day, and exposure to nephrotoxic drugs.

Obesity has been recognized as a risk factor for AKI in critically ill and septic patients, as reported by Gameiro et al. [23] and Ahn et al. [24], as well as in those undergoing cardiac surgery, according to Billings et al. [25]. The connection between BMI and AKI has been attributed to the role of adipose tissue in producing inflammatory mediators such as adipokines and leptin, while simultaneously reducing adiponectin levels in response to acute illness. These changes heighten the risk of developing AKI [26].

Our findings suggest that these same physiopathological pathways may promote the extension of the initial lesion, contributing to AKD.

HCT-CI index is a well-established risk predictor for overall mortality and relapse in patients undergoing hematopoietic cell transplantation. The independent association of an HCT-CI score equal to or above two points and AKD emphasizes the clinical relevance of this score from a nephrological perspective. This association was expected considering that many variables included in this index have been associated with kidney disease in various clinical settings (such as diabetes, obesity, cardiac disease, and sepsis).

The pathogenic role of the C-reactive protein in AKI has recently gained increasing recognition. This protein is now known not only as an inflammation biomarker but also as an important mediator contributing to the pathogenesis of several diseases related to inflammation [27,28]. Considering the kidney, the C-reactive protein activates the mitogen-activated protein kinase pathway and plays a key role in recruiting leukocytes into inflammatory sites in human renal distal tubular cells [29] promoting local and systemic inflammation. Animal models have concluded that C-reactive protein exacerbates ischemia-reperfusion injury-induced AKI [30]. By inhibiting the proliferation of damaged tubular epithelial cells and promoting fibrosis in injured renal tissue, C-reactive protein contributes not only to the initiation of AKI but also to its progression toward more severe and chronic stages [31].

Considering the several studies on AKI in HSCT, whether AKI is significantly more common in Allogeneic compared to autologous HSCT is a matter of contention. Still, the incidence of moderate-to-severe AKI tends to be significantly higher in allogeneic HSCT [32,33]. Moderate-to-severe AKI is associated with higher levels of inflammation, damage, and risk of renal injury progression. Considering this last aspect, we were not surprised to find an association between allogeneic transplantation and AKD.

Nephrotoxic drugs (including gentamicin, amikacin, vancomycin, amphotericin B, and foscarnet) were also associated with AKD in our study. The association of these drugs was already well established for AKI in HSCT [34,35,36,37]. The role of nephrotoxic drugs is also well known in other AKI scenarios; however, it has not been thoroughly studied for AKD. Our results suggest that nephrotoxic drugs may play a vital role in AKD. Unfortunately, it was not possible to discriminate the contribution of each nephrotoxic because patients were often on two or more of these drugs at the same time.

Although we were expecting CKD prior to HSCT to be a risk factor for AKD, we did not find that association in our study. We believe that the low incidence of CKD in our population (6.4%), combined with the fact that twenty-five out of the twenty-seven patients with CKD were patients with multiple myeloma who were undergoing autologous HSCT (already a subset of the population with lower AKD as mentioned above), did not allow us to infer such association. More representability of CKD in this population is necessary.

The impact of AKI on overall survival in HSCT patients has been studied by several authors in the last decade and has not been verified in some studies [38]. Many authors found lower overall survival associated only with severe stages of AKI [7,9]. In our study, although both AKI and AKD showed an association with overall survival in univariable analysis, only AKD persisted in the final multivariable model, conferring a 1.75-fold higher risk of death than patients with no AKD. Our results show the negative impact of AKD in 3-year overall survival in HSCT patients, superimposing AKI impact on this outcome. In other clinical settings, studies comparing AKI patients with delayed renal recovery with AKI patients with complete renal recovery showed higher mortality and major adverse cardiovascular events [39,40]. Therefore, enhancing preventive strategies, close monitoring, and early intervention for AKI in these patients is essential to reduce the risk of progression to AKD, which is linked to a poorer prognosis.

The primary limitation of this study stems from its retrospective design and single-center setting, which may restrict the generalizability of the findings. Additionally, due to data constraints, many patients were exposed to multiple nephrotoxic drugs, making it challenging to determine the individual effect of each medication on AKD. Furthermore, information on structural kidney abnormalities and proteinuria was not available.

Despite these limitations, our study has strengths worth mentioning. To the best of our knowledge, this is the first study on AKD in an HSCT population to take into account both creatinine and urinary output variations, and that considers death as a competing event, which we believe confers a more accurate diagnosis. Also, it is the first study on AKD in HSCT to consider a follow-up period longer than 100 days, demonstrating the long-term impact of AKD on the overall survival of this population for the first time.

Prospective and multicentre studies are needed to study AKD in this growing population in order to provide better care to our patients.

## 5. Conclusions

Our study shows the negative impact of AKD in 3-year overall survival in HSCT patients, superimposing AKI impact on this outcome. Therefore, it is necessary to reinforce preventive measures, monitoring and establish an early approach to AKI in these patients to prevent progression to AKD which is ultimately associated with a worse prognosis.

## Figures and Tables

**Figure 1 cancers-17-00538-f001:**
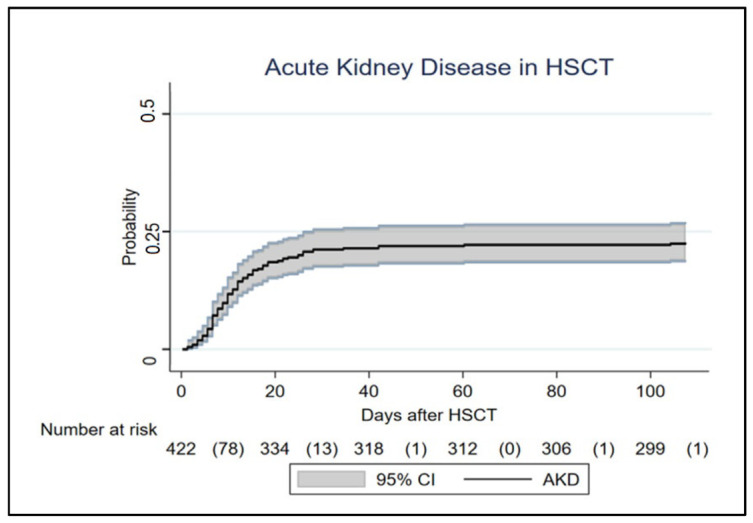
AKD cumulative incidence function in the first 100 days after HSCT. Death was considered a competing event. HSCT—hematopoietic stem cell transplant; CI—cumulative incidence; AKD—acute kidney disease.

**Figure 2 cancers-17-00538-f002:**
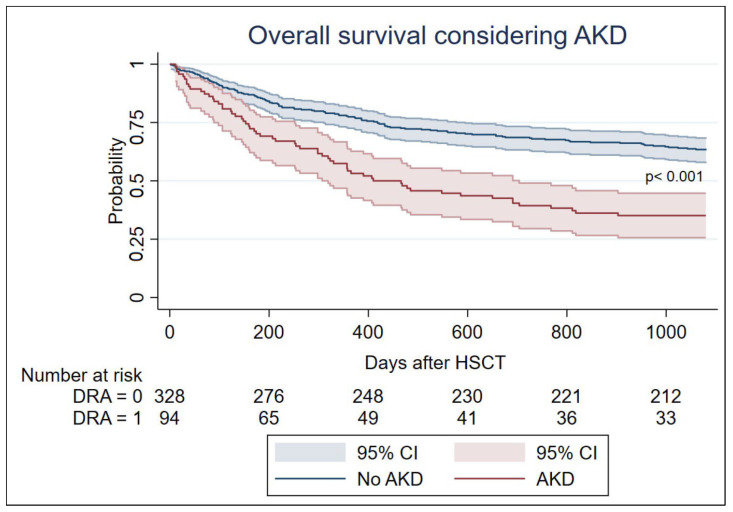
Overall survival considering AKD in the first 100 days after HSCT. HSCT—hematopoietic stem cell transplant; CI—cumulative incidence; AKD—acute kidney disease.

**Table 1 cancers-17-00538-t001:** Patients’ baseline characteristics and transplant-related variables.

Patients’ Characteristics	Category	n (%)	P50 (P25–75)
Age at transplant (years)			50.2 (36.0–59.5)
Gender	Male	236 (55.9)	
Race	Caucasian	385 (91.4)	
BMI (Kg/m^2^)			24.6 (21.9–27.8)
HCT-CI	0–1	353 (84.3)	
	≥2	66 (15.8)	
Hypertension		88 (20.9)	
Diabetes Mellitus		26 (6.2)	
Chronic Kidney Disease		27 (6.4)	
Hematologic Diagnosis	Leukaemia	164 (38.8)	
	Lymphoma	115 (27.3)	
	Multiple Myeloma	143 (33.9)	
Type of HSCT	Autologous	258 (61.1)	
	Allogeneic	164 (38.9)	
Type of donor	Self	258 (61.1)	
	Related	92 (21.8)	
	Not related	72 (17.1)	
Previous radiotherapy	yes	82 (19.5)	
Basal eGFR (mL/min/1.73 m^2^)			107.3 (94.3–122.1)
Conditioning Regimen	Myeloablative	305 (72.2)	
	Non-myeloablative	117 (27.8)	
Graft source	Peripheral Blood	389 (92.2)	
	Bone Marrow	33 (7.8)	
GVHD prophylaxis	CsA + MMF	117 (27.7)	
	CsA + MTX	47 (11.1)	
	None	258 (61.1)	
Acute GVHD		117 (27.7)	
Sepsis		135 (32.0)	
Nephrotoxic drugs		312 (74.1)	
Mucositis		202 (48.1)	
TMA/TLS/VOOS		15 (3.6)	
Shock		53 (12.6)	
Hemoglobin (gr/dL)			11.6 (10.2–12.6)
Leukocytes (cells/mm^3^)			4920 (3500–6860)
Neutrophils (cells/mm^3^)			2960 (1850–4420)
Platelets (/μL)			179,000 (127,000–245,000)
Uric Acid (mg/dL)			5.2 (4.0–6.0)
Reactive C Protein (mg/dL)			0.49 (0.15–2.00)
Lactate Dehydrogenase (U/L)			339 (291–426)
Albumin (mg/dL)			4.0 (3.7–4.5)

P50—median; P25—25th percentile; P75—75th percentile; BMI—body mass index; HCT-CI—hematopoietic stem cell transplant comorbidity index; eGFR—estimated glomerular filtration rate; CsA—cyclosporine; MMF—mycophenolate mofetil; MTX—methotrexate; TMA/TLS/VOOS—thrombotic microangiopathy/tumor lysis syndrome/sinusoidal obstruction syndrome.

**Table 2 cancers-17-00538-t002:** Competing risks regression. Univariable analysis for AKD.

Patient’s Characteristics	Hazard Ratio Estimate	95% Confidence Interval		*p*-Value
		Lower Limit	Upper Limit	
Age at Transplant (years)	1.00	0.99	1.01	0.748
Gender (reference category female)	0.81	0.54	1.21	0.301
Race (reference category caucasian)	1.02	0.50	2.09	0.949
BMI (Kg/m^2^)	1.04	1.00	1.08	0.036
HCT-CI (reference category ≥ 2)	1.65	1.02	2.67	0.040
Hypertension	1.70	1.10	2.63	0.016
Diabetes Mellitus	1.07	0.47	2.41	0.870
Chronic Kidney Disease	2.14	1.08	4.23	0.029
Type of HSCT (reference category allogeneic)	1.79	1.20	2.70	0.004
Previous Radiotherapy	0.81	0.48	1.38	0.442
Conditioning Regimen (reference category myeloablative)	1.29	0.98	1.87	0.260
Cell Source (reference category bone marrow)	0.70	0.37	1.32	0.264
GVHD prophylaxis (reference category MTX)	1.27	0.84	1.95	0.260
Acute GVHD	1.04	0.70	1.54	0.850
Shock	2.14	1.31	3.50	0.002
Nephrotoxic Drugs	8.33	1.11	62.66	0.039
Sepsis	1.53	1.02	2.30	0.042
Mucositis	1.09	0.73	1.63	0.676
TMA/TLS/VOOS	1.74	0.67	4.48	0.253
Hemoglobin (gr/dL)	0.97	0.87	1.09	0.638
Leukocytes (cells/mm^3^)	1.00	0.99	1.00	0.777
Neutrophils (cells/mm^3^)	1.00	0.99	1.00	0.859
Platelets (/μL)	0.99	0.99	1.00	0.592
Uric Acid (mg/dL)	0.99	0.92	1.05	0.554
C-Reactive Protein (mg/dL)	1.01	1.00	1.01	<0.001
Lactate Dehydrogenase (U/L)	1.00	0.999	1.00	0.656
Albumin (mg/dL)	1.00	0.94	1.06	0.966

BMI—body mass index; HCT-CI—hematopoietic stem cell transplant comorbidity index; GVHD—graft versus host disease; MTX—methotrexate; TMA/TLS/VOOS—thrombotic microangiopathy/tumor lysis syndrome/sinusoidal obstruction syndrome.

**Table 3 cancers-17-00538-t003:** Competing risks multivariable regression analysis for AKD.

Patient’s Characteristics	Hazard Ratio Estimate	95% Confidence Interval		*p*-Value
		Lower Limit	Upper Limit	
BMI	1.05	1.01	1.10	0.034
HCT-CI (reference category ≥ 2)	1.83	1.11	3.13	0.027
Type of Transplant (reference category allogeneic)	2.03	1.26	3.33	0.004
C-Reactive Protein (mg/dL)	1.01	1.01	1.02	<0.001
Nephrotoxic Drugs	4.81	1.54	4.95	0.008

BMI—body mass index; HCT-CI—hematopoietic stem cell transplant comorbidity index.

**Table 4 cancers-17-00538-t004:** Competing risks regression. Univariable analysis for death of all causes.

Patient’s Characteristics	Hazard Ratio Estimate	95% Confidence Interval		*p*-Value
		Lower Limit	Upper Limit	
Age at Transplant (years)	0.99	0.98	1.00	0.073
Gender (reference category female)	0.90	0.67	1.20	0.490
Race (reference category caucasian)	0.92	0.55	1.54	0.747
BMI (kg/m^2^)	0.94	0.91	0.97	0.001
HCT-CI (reference category ≥ 2)	1.18	0.80	1.73	0.398
Hypertension	1.56	1.05	2.32	0.029
Diabetes Mellitus	0.62	0.30	1.25	0.181
Chronic Kidney Disease	0.66	0.32	1.33	0.244
Type of HSCT (reference category allogeneic)	3.17	2.36	4.27	<0.001
Previous radiotherapy	0.79	0.54	1.17	0.252
Induction Regimen (reference category myeloablative)	1.54	1.12	1.73	0.101
Cells source (reference category bone marrow)	0.80	0.48	1.33	0.390
AKI	1.68	1.23	2.30	0.001
AKD	2.27	1.67	3.09	<0.001
Shock	7.47	5.31	10.51	<0.001
Nephrotoxic drugs	1.53	0.67	3.51	0.311
Sepsis	1.63	1.21	2.20	0.001
Mucositis	0.77	0.57	1.03	0.077
TMA/TLS/VOOS	1.46	0.72	2.98	0.291
Relapse	2.21	1.65	2.96	<0.001
Hemoglobin (gr/dL)	1.03	0.94	1.12	0.560
Leukocytes (cells/mm^3^)	1.00	0.99	1.00	0.280
Neutrophils (cells/mm^3^)	0.99	0.99	1.00	0.117
Platelets (/μL)	0.99	0.99	0.99	<0.001
Uric Acid (mg/dL)	1.10	0.97	1.25	0.130
Reactive C Protein (mg/dL)	1.00	0.99	1.01	0.180
Lactate Dehydrogenase (U/L)	1.62	1.37	1.87	<0.001
Albumin (mg/dL)	1.12	0.86	1.46	0.400

BMI—body mass index; HCT-CI—hematopoietic stem cell transplant comorbidity index; AKI—acute kidney injury; AKD—acute kidney disease; TMA/TLS/VOOS—thrombotic microangiopathy/tumor lysis syndrome/sinusoidal obstruction syndrome.

**Table 5 cancers-17-00538-t005:** Cox proportional hazards model regression. Multivariable analysis for mortality.

Patient’s Characteristics	Hazard Ratio Estimate	95% Confidence Interval		*p*-Value
		Lower Limit	Upper Limit	
AKD	1.75	1.27	2.39	0.001
Sepsis during hospital admission for HSCT	1.49	1.10	2.02	0.011
Type of transplant (reference category allogeneic)	3.33	2.43	4.54	<0.001
Relapse	2.61	1.93	3.54	<0.001
LDH (considering 100 units/L increments)	1.10	1.03	1.80	0.001

AKD—acute kidney disease; HSCT—hematopoietic stem cell transplant; LDH–serum lactate dehydrogenase.

## Data Availability

The data underlying this article will be shared on reasonable request to the corresponding author.

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
