# Peer review of "Risk Factors and Long-Term Outcomes of Acute Kidney Disease in Hematopoietic Stem Cell Transplant—Cohort Study"

_cancers, 2025, doi:10.3390/cancers17030538_

Round 1

Reviewer 1 Report

Comments and Suggestions for Authors

This article addresses an interesting topic from both a clinical care perspective and in terms of the potential long-term toxicities faced by patients undergoing hematopoietic stem cell transplantation (HSCT). 

Major Suggestions 

The primary suggestion I would make is that the study should separate the populations of autologous and allogeneic HSCT. Given the significant differences between these two procedures, it is highly relevant to understand the incidence of this complication and the associated risk factors in each group. For example, the use of nephrotoxic drugs is far more frequent in the allogeneic group, as is the incidence of both infectious and non-infectious complications that may compromise kidney function. Additionally, the impact on mortality might differ, being less pronounced in the autologous group. Moreover, a significant proportion of patients in the autologous group have multiple myeloma, a condition that is often associated with altered or unstable kidney function due to the disease itself. 

Minor Suggestions 

  • Abstract: The abstract mentions that AKD has a significant impact on overall survival (OS), but specific numbers should be provided to objectively reflect this finding in the Abstract. 

  • Grammar/Style in Abstract should be reviewed: For example, the sentence: “Retrospective cohort study considering AKD when AKI was observed and the patient remained in KDIGO stage 1 or greater for ≥7-days, using creatinine or urinary output criteria.” could be rewritten for clarity. 

  • Line 68: The acronym CKD should be defined upon first use. 

  • MDS Inclusion: Were patients with myelodysplastic syndromes (MDS) included in the study? Are they grouped under leukemias? 

  • Table 1: 

  • Correct the label allogenic to allogeneic. 

  • Use consistent capitalization (e.g., change peripheral blood to Peripheral Blood). 

  • Nephrotoxic Drugs: A description of the nephrotoxic drugs considered would be useful for interpreting the results. For example, was acyclovir, which is commonly administered universally to all patients with positive HSV or VZV serology in both autologous and allogeneic transplants, included in the list? Similarly, were cyclosporine, tacrolimus, and MTX—standard drugs in the allogeneic setting—considered? Since these drugs are used by almost all patients, what kind of nephrotoxic drugs are you referring to when you talk about risk factors? 

  • Engraftment Syndrome: Was the incidence of engraftment syndrome in the autologous group analyzed, and its impact on AKD? 

  • Acute GVHD: 

  • Was the incidence of acute GVHD in the allogeneic group analyzed, and its impact on AKD? 

  • Was acute GVHD included as a risk factor for mortality, OS, and PFS? 

  • Progression from AKD to CKD: How many patients who develop AKD within the first 100 days post-transplant eventually progress to CKD?

Author Response

Dear Revisor,

Thank you very much for your comments, we are incredibly happy to know that you consider our work interesting from a clinical perspective, and we do hope that our results will ultimately translate into better care for our patients.

Considering your primary suggestion of separating the populations of autologous and allogeneic hematopoietic stem cell transplantation (HSCT):

We completely agree with your statement pointing out that these two procedures are different and are associated with different incidences of infectious and non-infectious complications. Still, when considering Acute Kidney Injury (AKI), many studies are not consistent in considering allogeneic transplant statistically associated with higher incidences of AKI.  

In fact, our results did not support different incidences according to transplant type. In our population, the cumulative incidence of AKI in patients with multiple myeloma undergoing autologous HSCT was 49.7%, the cumulative incidence of AKI in patients with lymphoma undergoing autologous HSCT was 63.7% and the cumulative incidence of AKI in patients with leukemia undergoing allogeneic HSCT was 63.4%. When separated only by transplant type, autologous HSCT had a cumulative incidence of AKI of 56.8% and allogeneic HSCT had a cumulative incidence of AKI of 63.7%, and no statistical significance was found amongst the groups.

The fact that we did not find significant differences in AKI incidences was precisely the reason we considered it relevant to include both autologous and allogeneic HSCT patients in the same analysis for Acute Kidney Disease (AKD). Following our multivariable analysis, we were able to identify that the type of transplant does indeed have an independent effect on the progression to AKD (beyond the influence of other variables such as nephrotoxicity and infection). 

Considering your observation that a significant proportion of patients in the autologous group have multiple myeloma, a condition that is often associated with altered or unstable kidney function due to the disease itself:

Although this is absolutely true, we found exactly the opposite in AKI incidence (as mentioned above). We believe that the fact that patients with multiple myeloma had lower risk of developing AKI compared to patients with either lymphoma or leukemia may be related to lower treatment burden previous to HSCTs in patients with multiple myeloma, where HSCT is part of first-line treatment and is performed as consolidation after a good response to therapy. Patients with lymphoma or leukemia were often treated with nephrotoxic high-dose chemotherapy regimens before admission to HSCT, and, in these patients, HSCT is often used at relapse despite previous treatments. So, although myeloma patients are usually considered to be a high-risk population for AKI compared to other hematologic malignancies, this may not necessarily be the case during hospitalization for an autograft — a setting where at least a partial response to pre-transplant therapy is mandatory, reducing the effect of intrinsic unstable kidney function that manifests at diagnosis as a result of active disease.

Considering minor revisions:

Abstract: The abstract mentions that AKD has a significant impact on overall survival (OS), but specific numbers should be provided to objectively reflect this finding in the Abstract. 

Thank you for your remark, we included specific numbers (HR, CI, p) in the abstract for CI of AKD, for risk factors and for the impact of AKD in OS.

Grammar/Style in Abstract should be reviewed: For example, the sentence: “Retrospective cohort study considering AKD when AKI was observed and the patient remained in KDIGO stage 1 or greater for ≥7-days, using creatinine or urinary output criteria.” could be rewritten for clarity. 

We rephrased the sentence to:  “Retrospective cohort study was conducted, considering AKD when AKI was present, and the patient continued to meet the KDIGO criteria (creatinine and/or urinary output criteria) for 7 days or more.“ 

Line 68: The acronym CKD should be defined upon first use. 

Thank you very much, we corrected the text.

MDS Inclusion: Were patients with myelodysplastic syndromes (MDS) included in the study? Are they grouped under leukemias? 

Our population included four patients with MDS, and we grouped under leukemias.

Table 1: Correct the label allogenic to allogeneic. Use consistent capitalization (e.g., change peripheral blood to Peripheral Blood). 

We apologize for these details, thank you for pointing them out. We corrected the table.

Nephrotoxic Drugs: A description of the nephrotoxic drugs considered would be useful for interpreting the results. For example, was acyclovir, which is commonly administered universally to all patients with positive HSV or VZV serology in both autologous and allogeneic transplants, included in the list? Similarly, were cyclosporine, tacrolimus, and MTX—standard drugs in the allogeneic setting—considered? Since these drugs are used by almost all patients, what kind of nephrotoxic drugs are you referring to when you talk about risk factors? 

As we mentioned in Materials and Methods section, Definitions subsection, Nephrotoxic drugs included gentamicin, amikacin, vancomycin, amphotericin B and foscarnet. 

Yes, acyclovir is also a nephrotoxic drug, but we did not include it considering that all patients were under this medication. Considering GVHD prophylaxis, for institutional reasons the only calcineurin inhibitor used was cyclosporine and unfortunately, we did not have access to serum concentrations of the drug , so we did not include it. The second drug for this purpose was either MMF (mycophenolate mofetil) or MTX (methotrexate). Because some studies have pointed MTX as a possible risk factor for AKI in HSCT, we considered the variable “GVHD prophylaxis” using as reference category patients on MTX in table 2 (by doing so, we did not include it in the category of nephrotoxic drugs). 

Engraftment Syndrome: Was the incidence of engraftment syndrome in the autologous group analyzed, and its impact on AKD? 

We did not analyze the incidence of engraftment syndrome (ES). We consider it an important complication of HSCT, AKI has been recognized in ES, is one of Spitzer’s minor criteria for diagnosis. Thus, it would be of clinical interest to study its relation to AKD. Unfortunately, the retrospective nature of our study did not allow us to collect this variable with accuracy. ES was not identified/excluded in the medical records, so the variable would have to be collected considering clinical descriptions that were often incomplete.

Acute GVHD: Was the incidence of acute GVHD in the allogeneic group analyzed, and its impact on AKD? Was acute GVHD included as a risk factor for mortality, OS, and PFS? 

Yes. Acute GVHD occurred in 117 patients, translating in an incidence of acute GVHD of 27,7% (in allogeneic HSCT of 71.3%). The univariable analysis for AKD showed no statistically significant impact with an HR estimate of 1.04 (0.70-1.54; p = 0.850). Initially we did not include in the table, but we agree it is an important result to share considering the association to AKI found in previous studies. We added this information to both table 1 and table 2.

Progression from AKD to CKD: How many patients who develop AKD within the first 100 days post-transplant eventually progress to CKD? 

Thank you very much for your comment on this matter, it is certainly a very relevant point which we had not initially approached. Considering the 94 patients that developed AKD within the first 100 days post-transplant, 26 died in the first 3 months after the AKD diagnosis, thus not reaching the timeline for CKD. Amongst the 68 patients that survived more than 3 months after the AKD diagnosis, 19.1% (13 patients) progressed to CKD. We inserted this information in the results.

Reviewer 2 Report

Comments and Suggestions for Authors

The manuscript by Natacha Rodrigues and co-workers describes an examination of Acute Kidney Disease (AKD) in Hematopoietic Stem Cell Transplant (HSCT) patients. The research fills a significant knowledge gap in the field, as AKD has been understudied compared to Acute Kidney Injury (AKI) in this population. The large sample size of 422 patients provides robust statistics. Also, the important inclusion of both creatinine and urinary output criteria for AKD diagnosis enhances diagnostic accuracy. The authors demonstrate that AKD is associated with a higher mortality risk and shows stronger prognostic value than AKI for overall survival. The manuscript provides clinically relevant risk factors for patient stratification. The findings are supported by the data and will be practically useful for HSCT patient care. The three-year follow-up period for survival analysis makes the results especially valuable. The manuscript deserves publication.

I have the following minor comments.

Figure 1:  The upper part of the plot is empty. Hence, it is reasonable to rescale the ordinate axis and make a break on this axis somewhere between 0.25 and 1.0.

The supplementary file containing the Excel table was submitted along with the main manuscript. However, this table is not cited in the manuscript, nor is it mentioned in the Supplementary section (this section is absent). 

Text in Portugese language is given in the supplementary table. This should be translated into English.

Summarizing, I recommend acceptance of the manuscript for publication after minor revision.

Author Response

Dear Revisor,

Thank you so much for your validation!

We are grateful for your suggestions and have proceeded accordingly.

Please ignore the supplementary file, I mistakenly put a file that was not intended to be there.

Round 2

Reviewer 1 Report

Comments and Suggestions for Authors

The authors' answers to the questions seem to me to be adequate. I therefore consider that the article can now be accepted.